# Geodemographic Area Classification and Association with Mortality: An Ecological Study of Small Areas of Cyprus

**DOI:** 10.3390/ijerph16162927

**Published:** 2019-08-15

**Authors:** Demetris Lamnisos, Nicos Middleton, Nikoletta Kyprianou, Michael A. Talias

**Affiliations:** 1Department of Health Sciences, School of Sciences, European University Cyprus, Nicosia 1516, Cyprus; 2Department of Nursing, School of Health Sciences, Cyprus University of Technology, Limassol 3041, Cyprus; 3Healthcare Management Postgraduate Program, Open University Cyprus, P.O. Box 12794, Nicosia 2252, Cyprus

**Keywords:** health inequalities, mortality, area classification, cluster analysis, geodemographics, Poisson spatial model, Cyprus

## Abstract

Geographical investigations are a core function of public health monitoring, providing the foundation for resource allocation and policies for reducing health inequalities. The aim of this study was to develop geodemographic area classification based on several area-level indicators and to explore the extent of geographical inequalities in mortality. A series of 19 area-level socioeconomic indicators were used from the 2011 national population census. After normalization and standardization of the geographically smoothed indicators, the k-means cluster algorithm was implemented to classify communities into groups based on similar characteristics. The association between geodemographic area classification and the spatial distribution of mortality was estimated in Poisson log-linear spatial models. The k-means algorithm resulted in four distinct clusters of areas. The most characteristic distinction was between the ageing, socially isolated, and resource-scarce rural communities versus metropolitan areas with younger population, higher educational attainment, and professional occupations. By comparison to metropolitan areas, premature mortality appeared to be 44% (95% Credible Intervals [CrI] of Rate Ratio (RR): 1.06–1.91) higher in traditional rural areas and 36% (95% CrI of RR: 1.13–1.62) higher in young semi-rural areas. These findings warrant future epidemiological studies investigating various causes of the urban-rural differences in premature mortality and implementation policies to reduce the mortality gap between urban and rural areas.

## 1. Introduction

Geographical inequalities in health refer to observed differences in population health across geographical areas. Area differences in the health status of populations have long been established in the literature, and numerous research studies have consistently shown that people in different geographical areas experience different degrees of ill health [1,2,3]. Even though the magnitude of geographical health inequalities varies across countries, the universality of the phenomenon was best described by Johan Mackenbach [4], to quote: “*… even if data for a particular country are not available, one can confidently expect similar inequalities to exist there as well*”. Cyprus is one such example of a country where the magnitude of geographical health inequalities is not known.

Geographical health inequalities are the result of differential opportunities, assets, resources, and constraints, which may be largely determined by upstream factors [3,5,6]. Several features of the physical and social environment of an area may impact on population health through various mechanisms [7]. In their conceptual model, Diez Roux and Mair [8] attribute the geographical disparities in health both to the unequal distribution of resources and to the unavoidable differences in the sociodemographic composition of communities. 

Studies commonly investigate the socioeconomic determinants of geographical inequalities in health by exploring the extent to which area-level indices of socioeconomic deprivation or disadvantage explain the observed geographical inequalities. These indices are constructed by combining several census or other routine data-based area-level indicators as proxies of the socio-economic environment of a community, such as level of educational attainment, employment statistics, and housing tenure [3]. Principal components analysis (PCA) or factor analysis are commonly used to combine this information into a composite index in order to place communities on the socioeconomic disadvantage continuum [9,10,11,12,13,14,15,16,17]. These deprivation indices are then used to quantify the magnitude of geographically-determined social inequalities in health. Although PCA and factor analysis approaches are helpful in revealing underlying dimensions of the latent construct of socio-economic disadvantage, they assume that all areas are homogeneous in possessing these area dimensions. As such, they ignore the possibility that there might exist a diversity of areas with different combinations of these dimensions [18,19]. For example, disadvantaged neighborhoods can be characterized by high poverty and high residential mobility but can also be characterized by high poverty and low residential mobility [20]. Furthermore, aggregated measures from PCA and factor analysis cannot identify the distinct characteristics, or combination of characteristics, attributable to specific geographical areas which may explain the observed variations in population health [21]. 

As an alternative to composite measures of deprivation, area classification approaches have also been proposed in order to classify small geographical areas into clusters, or groups of areas, on the basis of the similarity of their characteristics [18,22,23]. Using a range of area-level socio-demographic and socio-economic indicators from the census and/or other routine data sources, this approach classifies communities by the predominant characteristics of the locality and its residents [24,25]. The resulting clusters are internally homogeneous but different from other clusters. Based on their distinct profile, these clusters are generally given names which best reflect their defining characteristics as compared to other clusters. Thus, the aim of this approach is not to construct an ordinal measure of relative socio-economic disadvantage but to develop a descriptive classification system in which areas sharing similar distinctive patterns of sociodemographic, economic, housing or environmental features are grouped together. This grouping may capture an underlying diversity in community environments which is masked by deprivation indices [26]. Geodemographic classifications are increasingly used to investigate geographical health inequalities [27,28,29], describe variations in health service use [23,30], target communities in public health campaigns [31], or design tailored interventions [32].

In Cyprus, there is neither a commonly accepted index of area deprivation nor any geodemographic area classification. Thus, the first objective of this study was to develop and map for the first time a geodemographic area classification based on several area-level indicators derived from the latest national census of 2011. The second objective was to explore the extent of geographical inequalities in adult all-cause premature mortality (ages 15–65) and all-cause mortality in adults over 65 years of age. This is the first study to reveal the underlying ecological structure and diversity of Cypriot communities based on their socioeconomic characteristics.

## 2. Materials and Methods 

### 2.1. Data

Demographic and socioeconomic data were obtained from the 2011 national population census, conducted by the National Statistical Service. Individual data were aggregated at the level of geographical units used for census purposes. There are N = 369 communities (rural areas) and municipalities (metropolitan areas) in the areas controlled by the Republic of Cyprus. With a median population of 316 people (IQR: 111–1124, only 10% of areas >4100), these areas are generally very small by comparison to similar studies in the published literature due to the small size of the island. A series of 19 indicators were available from five main domains: demographic structure (e.g., proportion of retired population), household composition (e.g., proportion of single-parent households), housing features (e.g., proportion of dwellings that are vacant or for demolition), socio-economic status of the residents (e.g., proportion of population with low educational attainment and proportion of not owner occupied households), and occupational structure (e.g., proportion of population in elementary occupations) (Table 1). 

The number of deaths per gender and age-group with 5-year age bands starting from 15 to 20 years old for the period 2009–2011 and each geographical census area were obtained from the Health Monitoring Unit of the Cyprus Ministry of Health.

The study protocol for the investigation into the geographical patterning of cancer and all-cause mortality in Cyprus and its association with area-level socioeconomic characteristics was waived from further evaluation by the Cyprus National Bioethics Committee (EEBK EP2012.01.24). The data were provided upon official request by the Cyprus Statistical Services and the Health Monitoring Unit, Cyprus Ministry of Health, after approval by the Cyprus Ministry of Health Research Promotion Committee (5.34.01.7.2E). Furthermore, the study received exemption from the Commissioner for the Protection of Personal Data (3.28.33) since aggregated statistical data were used and no personal identifiers could be extracted.

### 2.2. Statistical Methods

Socio-demographic indicators exhibit spatial autocorrelation since adjacent geographical areas are more likely to display similar socio-economic characteristics. Furthermore, there is varying precision in the estimates across the various communities, with very low precision in the least dense rural and remote communities due to the particularly small underlying population denominators. Therefore, a univariate Gaussian spatial model was used to incorporate spatial dependence and to provide smooth estimates of each census indicator for each community. This model includes three terms, the constant term representing the national mean of the census indicator, the spatially structure random effect introducing spatial correlation in the data and the normally distributed random error [17]. Spatial correlation in random effects was incorporated by the conditional autoregressive (CAR) model [33], and the first order adjacency was used to define a community’s neighbors (i.e., those communities with a common border). The percentage of spatially structured variability to the total variability in each indicator assessed the geographical patterning in each indicator and its tendency to geographically cluster. With the exception of four indicators (namely, proportion of population with at most lower secondary education, proportion of population in elementary occupations, proportion of single-person households, and proportion of households without access to a personal computer), there was a need to transform all the rest using the logarithmic transformation log(1+x) in order to normalize their distribution. The indicators were normalized because clustering methods encounter problems with highly skewed distributions that are quite often observed in census data.

After normalization and standardization of the geographically smoothed indicators, the k-means cluster algorithm was implemented to classify communities with similar characteristics [34]. The objective of the k-means algorithm is to minimize the within-cluster variability. If the number of clusters within the data set is specified, the k-means classifier forms the clusters that are as distinct from each other as possible [35]. The number of clusters was chosen by combining the results of the steepest increase method and homogeneous size of clusters method [24]. In the steepest increase method, the average within-cluster sum of squares is found for different number of clusters *k* using the k-means algorithm. This average within-cluster sum of squares is then plotted against the number of clusters *k* and a kink in the curve indicates the optimal number of clusters. The homogeneous size of clusters method computes the average squared difference between the number of members in each cluster from the mean (the mean is the optimal solution as all clusters will have the same number of members) for different number of clusters *k*. The optimal number of clusters for this method is the one minimizing the average squared difference.

Poisson log-linear spatial models were used to estimate the spatially smoothed rate ratio of mortality across the Cypriot communities and municipalities and to investigate the differences in all-cause premature adult mortality (ages 15–65) and all-cause mortality for adults over the age of 65 across the geodemographic classification of areas. The dependent variable in these models was the observed numbers of adult premature mortality or mortality in over 65s. The offset term was the expected number of deaths in each geographical area and was estimated using indirect standardization, while the spatial random effect that incorporated spatial correlation followed the CAR model with first order adjacency. In the case of the investigation of differences in mortality across the geodemographic area classification, the independent variable was the identified geodemographic area classification index, which was entered in the model as *k* − 1 dummy variables. The univariate Gaussian spatial model and the Poisson log-linear spatial model were implemented in a Bayesian setting, which requires a specification of the prior distributions for the regression coefficients and the variance components. A multivariate Gaussian prior was used for the regression coefficients with constant zero-mean vector and diagonal elements of variance matrix equal to 100,000. The variance parameters of the random error and the CAR model were assigned inverse-gamma prior distributions with shape equal to 1 and scale equal to 0.01. All statistical analyses were performed in the statistical software R [36] and WinBUGS [37]. The CARBayes package was used for fitting the Poisson log-linear spatial model [38]. Credible intervals of 95% were computed for the spatial models.

## 3. Results

The majority of the census indicators exhibited geographical patterning. Spatial clustering was particularly marked for the proportion of the population aged 0–14 and aged over 65, for low educational attainment, no PC in household, six and over member households, agricultural workers, and secondary/seasonal households (see Appendix A). The k-means cluster algorithm in combination with the steepest increase method and homogeneous size of clusters method suggested four distinct clusters of communities. Figure 1 presents a choropleth map of the four geodemographic clusters identified across the 369 Cypriot communities, while Figure 2 displays the radial plots summarizing the mean of each census indicator for each cluster. 

The most characteristic distinction was between the ageing, socially isolated, and resource-scarce rural communities versus metropolitan areas with a younger population, higher educational attainment, and professional occupations. The metropolitan areas include all the large cities but also extend to the surrounding and often suburban sprawling areas with easy commuting access to the nearest city. Rural areas were further sub-divided into three distinct clusters. The first cluster appears to geographically correspond to the mountainous areas in the center of the island. It was named “traditional-life rural communities”, as these communities are mainly agricultural-based economies with an ageing population, low educational attainment, and living in single-person households. The population in these areas is predominantly Cypriot. Furthermore, one defining characteristic is the higher proportion of secondary/seasonal houses, presumably for family members to visit relatives at weekends and holidays as it is customary in the Cypriot culture. The second cluster, named “young semi-rural areas” are communities with a mixed economy based on agricultural activities and elementary occupations, but also a mixed demographic composition. While there is a high proportion of retired population, these communities are living and working communities with large families and young children. Low educational attainment and lack of amenities (e.g., access to a PC) seems to be more common in both the traditional rural and young semi-rural areas. Interestingly though, by comparison to metropolitan areas, home ownership is relatively high while unemployment is relatively low in these generally more socioeconomically disadvantaged areas. High proportion of home ownership, however, does not capture the quality of the housing, while it is likely that unemployment figures do not capture the seasonal nature of agriculture and potential ‘hidden’ unemployment not recorded in official statistics. Finally, the fourth cluster included the communities on the west coast of Cyprus (i.e., the wider area of the Paphos district), not including the metropolitan areas in and around the city of Paphos. It seems that the most defining features of this cluster of generally rural areas are the higher percentage of non-Cypriot population, higher proportion of buildings constructed post 2000, and higher proportion of private renting. At the same time, these generally traditional communities appear to have a high proportion of dwellings that are vacant and for demolition. Thus, it was named “regenerated west coast communities”. In terms of socioeconomic disadvantage, while access to a PC is relatively high in sharp contrast to the rest of the rural areas on the island, there seems to be higher unemployment in these communities, possibly indicating the seasonal nature of employment in the construction sector and tourist industry (Figure 2).

The choropleth map of smoothed rate ratios (RR) of premature adult mortality is presented in Figure 3. The smoothed RR varied by almost two-fold (range of RR, 0.79–1.25) across the communities with higher rates in less dense, rural, and mountainous areas of Cyprus and lower rates in and around the capital city of Nicosia. Table 2 presents the relative risks of premature adult mortality across the four identified distinct clusters. There appeared to be a clear urban-rural divide in premature adult mortality, with rural areas at a disadvantage, irrespective of their socio-demographic profile. By comparison to metropolitan areas, premature mortality appeared to be 44% (95% CrI of RR: 1.06–1.91) higher in traditional rural areas and 36% (95% CrI of RR: 1.13–1.62) higher in young semi-rural areas. Premature adult mortality also appeared slightly elevated in the west coast communities, even though the observed difference was not statistically significant (RR = 1.17, 95% CrI: 0.97–1.39). 

In contrast to premature mortality, a different picture emerged with regards to mortality in adults over the age of 65. Rate ratios of all-cause mortality in adults aged over 65 are presented in Table 3. While rates of premature mortality were higher in traditional-life rural communities, these communities appear to have 12% lower mortality rates in adults aged over 65 compared to metropolitan areas (95% CrI for RR: 0.77–0.99). Mortality in adults aged over 65 also appeared slightly lower in young semi-rural communities (RR = 0.96, 95% CrI: 0.87–1.05) and west-coast communities (RR = 0.93, 95% CrI: 0.84–1.03), even though these differences were not statistically significant.

## 4. Discussion

This is the first study to explore the socioeconomic profile of communities in Cyprus and the extent that this profile explains inequalities in mortality. The geodemographic classification of 369 small-area Cypriot communities revealed four clusters. These clusters appear to have a clear geodemographic interpretation as well as a geographical structure, defined in terms of the proximity to large centers of population as well as geomorphological features, i.e., mountainous communities and communities on the west coast. The exploration of differential health outcomes across these four geodemographic clusters revealed clear urban-rural health inequalities. While not striking in terms of magnitude, it seems that premature adult mortality is 36–44% higher in the rural areas of the island. In contrast, mortality rates in adults aged over 65 appeared to be up to 12% lower, at least in the more traditional mountainous rural communities. Interestingly, mortality rates were not higher in the generally rural communities on the west coast of the island which are characterized by a higher influx of non-Cypriot population of retired age in recent decades.

Even though rural areas were further sub-divided in three distinct groups with a different socio-demographic profile, the most striking feature revealed in the analysis was the sharp distinction between the generally ageing, socially isolated, and resource-scarce rural communities versus the metropolitan areas with a younger population, in professional occupations, and higher educational attainment. Several other international geodemographic classifications produced clusters reflecting a continuum ranging from urban to most rural. Some examples are the 2011 area classification for output areas (2011 OAC) in the UK [25], the National Center for Health Statistics’ (NCHS) urban-rural classification scheme for counties in the USA [39], the geodemographic classification in Japan [40], and more recently, in Thailand [21]. 

It is interesting to note that in this study, a number of otherwise rural communities on the west coast of Cyprus formed a distinct cluster with more defining features the higher proportion of non-Cypriot population and the higher proportion of buildings constructed post 2000. Indeed, areas on the west coast of Cyprus experienced a development boom in recent decades. Consequently, new residential developments were in demand and the economic sector of construction experienced a large growth. This was a period when many European citizens bought properties in these areas, regenerating the otherwise rural and socioeconomically disadvantaged communities [41]. Nevertheless, this sector was one of the first hit by the economic crisis of the years 2011–2013, possibly explaining the higher unemployment rate as recorded in official statistics.

There appeared to be geographical health inequalities on the small island of Cyprus, with rural areas at a disadvantage in terms of premature adult mortality. In recent years, an increasing number of studies have investigated urban-rural inequalities in health outcomes and/or health-related adverse behaviors. Even though, the findings have generally been mixed [42,43], the assumption that rural populations are by comparison healthier is increasingly being challenged. A study in the USA examined sex-specific mortality rates for selected causes of death during the period of 2008–2010 across urbanization levels and found that mortality for adults aged 25–64 was lowest in the fringe counties of large metro areas and increased steadily as counties become more rural. The age-adjusted death rate in rural counties was 44% higher for males and 47% higher for females. A further study from the UK examined differences in mortality rates between rural and urban areas in England and Wales in the years 2002–2004 [44] and found that the observed mortality differences between rural and urban areas were largely explained by the differences in the distribution of socioeconomic circumstances. The inconsistencies in urban-rural comparisons in terms of health outcomes may be attributed to the various definitions of rurality in use by international studies as well as the fact that findings may be context-specific. Moreover, the nature of rurality itself varies within a country, as this study also suggests, across time and even more across countries [45,46].

The extent to which the elevated premature mortality in rural areas reflects the effect of the continuing decline in rural economy in Cyprus, and/or barriers to health care and services, and/or differences in lifestyle and behavior determinants of health, or is even restricted to certain causes is not known. Of course, the nature of ‘rurality’ itself may present a different profile across different parts of the country, as this study suggests. It is also likely that it reflects historical selective migration processes, which may be different across different parts of the country. For example, in terms of the more traditional small mountainous communities, it is likely that there has been selective out-migration of healthier individuals in search of employment opportunities. Such processes, and the extent to which they impact on mortality rates, have been described in the literature. For instance, in France, there is evidence to suggest that the largest increases in premature mortality in a 30 year period were seen in areas with the largest population decline, and in fact more so in socioeconomically deprived areas [47]. Unfortunately, the mortality registry in Cyprus only goes back to 2007, while there is no good historical information on population movement.

Rural economy in Cyprus has been experiencing a continuing decline while economic growth has been largely restricted to the main cities of the island, most likely leading to selective migration of healthier individuals. In parallel, other selective migration processes may be at play. For example, people may move back to rural communities after retirement, or in some cases, after early retirement due to underlying health problems. This selective migration process may explain the higher magnitude of geographical inequalities in premature adult mortality in comparison to mortality over 65 and the different direction of geographical inequalities between these two outcomes. Since this is a first-time exploratory ecological study into the urban-rural patterning of mortality, the extent to which such processes are at play and their potential effect in determining the observed urban-rural differences are not clear. A study in the UK used data from the Office for National Statistics Longitudinal Study (ONS LS) to assess the role of residential mobility within England between 1981 and 2001 in explaining geographic inequalities in all-cause mortality between urban and rural local authority districts at the end of the period (i.e., deaths occurring between 2001 and 2005) [48]. The study suggested that residential mobility in the period of 1981–2001 accounts for about 30% of the observed urban-rural inequalities in mortality. The study also suggested that members of working age (20–64 years) moving out of rural areas and members of retirement age (65 years and older) moving into rural areas were shown to be healthier. Interestingly, in the case of this study, in the west coast communities where there was a pattern of in-migration of retired population from European countries, there was no evidence of elevated mortality among people over 65, even though similarly to the rest of the rural areas of the island, the rate of premature mortality appeared slightly elevated compared to metropolitan areas.

In terms of the access to health care and other services, rural communities in Cyprus are at a clear disadvantage. There are very few primary health care centers in rural areas [49], while in recent years, a number have ceased to operate after a governmental decision to restructure the healthcare system for organizational efficiency [50]. There is considerable evidence in the literature supporting the idea that transportation barriers are an important barrier to healthcare access [51]. A cross-sectional study conducted in Cyprus and Greece between 2005 and 2009 found that long-term access to advanced healthcare services (i.e., both primary and secondary services rather than just primary units operated in rural areas) and nutrition (e.g., dieticians) services were correlated with decreased burdens of common cardiovascular risk factors amongst the elderly Cypriot population [52].

Major contributors of the elevated burden of morbidity and premature mortality afflicting rural communities in the USA were found to be obesity and insufficient physical activity [53,54]. A cross-sectional study conducted in Cyprus in the year 2009 using a representative sample of adults aged 24–65 years found that individuals residing in rural areas had 30% higher likelihood of being obese compared to those living in urban areas [55]. 

No clear inferences can be drawn as to the actual mechanisms underlying the observed geo-socioeconomic patterning of premature mortality in Cyprus. Other than the fact that this is the first study to explore the spatial patterning of mortality, or any other health indicator, there is generally a lack of studies exploring rural health on the island. As discussed above, several mechanisms may underlie the observed pattern in premature mortality, and it is more likely that these factors are at play in parallel. To summarize, these mechanisms may be compositional or contextual or a combination of both [56]. Compositional factors refer to the characteristics of the population that lives there as a result of a process of selective migration (out-migration of healthier individuals and/or in-migration of less healthy individuals). Contextual factors refer to actual features and characteristics of these places which may reflect both the lack of access to good quality healthcare and prevention and health promotion services as well as the characteristics of the actual socio-cultural environment. 

A clear limitation of this study is the subjective decisions made in the process of running the cluster analysis [35]. This is because there are no set statistics that can objectively answer what precise decisions should be made at each stage of the analysis. To minimize the effect of decision on the results, any decisions taken in the process were informed by multiple methods to ensure that they were justified [24]. Furthermore, the data used to develop the geodemographic area classification was extracted from a single decennial census of the population. Such classifications are often criticized as becoming less useful over time because of the changing composition of small geographical areas [57]. Census geodemographics are by definition constrained to looking at patterns of sociodemographic composition and other characteristics of the community environment every 10 years. This is a critical constraint given how rapidly the socio-demographic compositions and socioeconomic characteristics of communities may change. In addition, this study was largely dependent on data availability from the Cyprus national census, which is not as rich compared to other European censuses. In fact, the lack of any other open data source of routine data at a small geographical level in Cyprus which could be integrated to census data is a clear limitation. Another limitation is the ecological design of the study, which does not allow any inferences in terms of the underlying determinants of the observed patterning of mortality and in particular in terms of the potential effect of selective migration [48]. Finally, the size and configuration of areal units can directly influence the results of any statistical analysis, which is known as the modifiable area unit problem [58]. Mitchell et al. [59] have emphasized the advantages that come from working with areas of relatively equal population size or social homogeneity. In this study, we have used the smaller geographical unit for which data was available from the 2011 National Population Census and the Health Monitoring Unit. Nevertheless, the study only looked at all-cause mortality, and in fact for a three-year period, since it was not feasible to look at cause-specific mortality at this small level of geographical aggregation.

Future efforts will be concentrated in extending the developed geodemographic classification to include data from other routine sources or even any useful survey data on local behaviors, beliefs, and/or habits, either opportunistically or from specially-designed surveys with national coverage, to provide a more robust understanding of the differences in the sociodemographic composition of Cypriot communities. Moreover, the extent to which the observed geographical health inequalities reflect the effect of a selective in and out migration process, and/or barriers to health care and services, and/or lifestyle and behavior determinants of health is not known. These possible causes of geographical health inequalities need to be examined in specially designed epidemiological studies.

## 5. Conclusions

This study developed and mapped geodemographic area classification for first time in Cyprus, which revealed a clear interpretable classification of communities into four clusters. The developed geodemographic classification will offer the potential to act as a catalyst for “spatial thinking” among local decision-makers, which is currently lacking in public health policy. Resource allocation of healthcare facilities and services in Cyprus has not traditionally followed a health need assessment-based approach. Allocation of health services and resources has been mainly driven by financial capacity. Nevertheless, the fact that the country is currently undergoing a major healthcare system reform with the introduction of a national health care system presents new opportunities. Thus, the understanding of the geo-socioeconomic patterning of mortality across Cypriot communities provided by this study is more relevant now than ever. The geographical health inequalities in terms of premature adult mortality revealed by this study warrant future epidemiological studies investigating various causes of the urban-rural differences in premature mortality and implementation policies to reduce the mortality gap between urban and rural areas.

## Figures and Tables

**Figure 1 ijerph-16-02927-f001:**
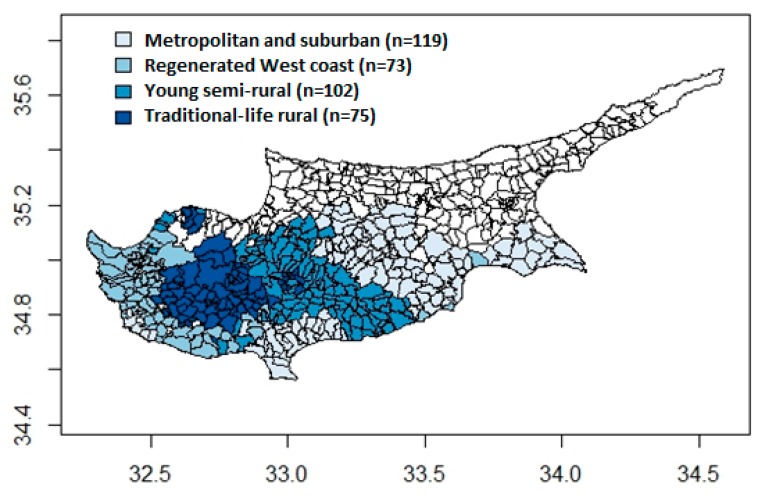
Choropleth map of the geodemographic area classification of 369 Cypriot communities and municipalities based on nineteen 2011 census-based sociodemographic indicators.

**Figure 2 ijerph-16-02927-f002:**
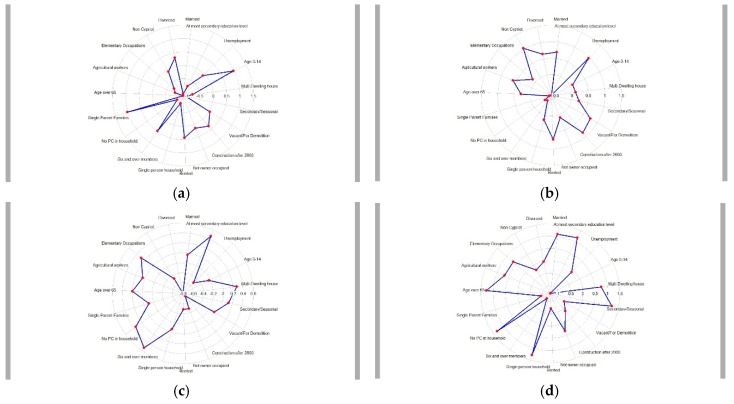
Radial plots displaying the mean of the standardized values of each census indicator for the four identified clusters of Cypriot communities. The numbers on the scale represent the difference from the mean value for that census indicator (the mean is represented by the ring at 0). (**a**) metropolitan and suburban areas; (**b**) regenerated west coast communities; (**c**) young semi-rural communities; (**d**) traditional-life rural communities.

**Figure 3 ijerph-16-02927-f003:**
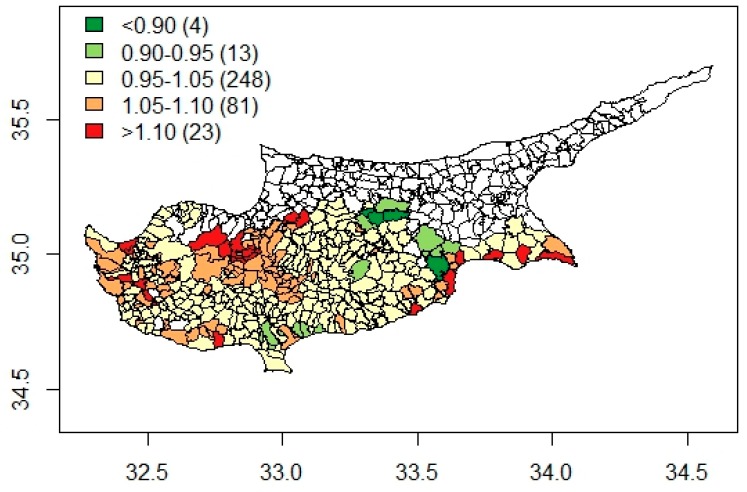
Choropleth map of smoothed rate ratios of all-cause premature adult mortality across 369 Cypriot communities and municipalities.

**Table 1 ijerph-16-02927-t001:** Definition of census indicators used to create the geodemographic area classification in Cyprus.

Indicator	Definition
**Demographic**	
Age 0–14	Population aged 0–14 years, as a proportion of the total population
Age over 65	Population aged over 65, as a proportion of the total population
Non-Cypriot population	Population identifying as non-Cypriot, as a proportion of the total population
Retired population	Retired population, as a proportion of the total population
**Household Composition**	
Married population	Married persons, as a proportion of the total population
Divorced population	Divorced persons, as a proportion of the total population
Single-person households	Households with a single person, as a proportion of total households in conventional dwelling
Single-parent households	Households with a single parent, as a proportion of total households in conventional dwelling
Six and over members	Households with six and over members, as a proportion of total households
**Housing features**	
Multi dwelling house	Multi dwelling houses, as a proportion of total conventional dwellings
Secondary/Seasonal	Secondary/seasonal living quarters, as a proportion of total living quarters
Vacant/For demolition	Vacant/for demolition living quarters, as a proportion of total living quarters
Construction after 2000	Conventional dwellings constructed after 2000, as a proportion of total conventional dwellings
**Socio-Economic Status**	
Not owner-occupied	Not owner-occupied households, as a proportion of households in conventional dwelling
Privately renting	Rented households, as a proportion of households in conventional dwelling
Educational attainment	Persons with at most lower secondary education, as a proportion of total population over 15 years of age
No PC in household	Households without access to a personal computer, as a proportion of total households
**Occupational Structure**	
Unemployment	Unemployed economically active persons, as a proportion of total economically active population
Agricultural workers	Persons in employment working in agriculture, as a proportion of total employed population
Elementary occupations	Persons in employment working in elementary occupations, as a proportion of total employed population

**Table 2 ijerph-16-02927-t002:** Posterior median and mean of rate ratios (RR) with 95% credible intervals of all-cause premature adult mortality (ages 15–65) across the geodemographic area classification.

Cluster	Median, Mean of RR (95% Credible Interval)
Metropolitan and suburban areas	Ref
Regenerated west coast communities	1.17, 1.16 (0.97, 1.39)
Young semi-rural communities	1.36, 1.37 (1.13, 1.62)
Traditional-life rural communities	1.44, 1.44 (1.06, 1.91)

**Table 3 ijerph-16-02927-t003:** Posterior median and mean of rate ratios (RR) with 95% credible intervals of all-cause mortality in adults aged 65 years and over across the geodemographic area classification.

Cluster	Median, Mean of RR (95% Credible Interval)
Metropolitan and suburban areas	Ref
Regenerated west coast communities	0.93, 0.94 (0.84, 1.03)
Young semi-rural communities	0.96, 0.95 (0.87, 1.05)
Traditional-life rural communities	0.88, 0.89 (0.77, 0.99)

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
