# Peer review of "Geodemographic Area Classification and Association with Mortality: An Ecological Study of Small Areas of Cyprus"

_ijerph, 2019, doi:10.3390/ijerph16162927_

Round 1

Reviewer 1 Report

Thank you for the opportunity to review the manuscript "Geodemographic Area Classification and Association with Mortality: An Ecological Study of Small Areas of Cyprus".

The main purpose of the manuscript was to develop and map for first time a geodemographic area classification based on several area-level indicators derived from the latest cipriot census of 2011, starting from the knowledge gap: "the  magnitude of geographical health inequalities in Cipru is not known" . The authors concluded that "the developed geodemographic classification will offer the potential to act as a catalyst for “spatial thinking” among local decision-makers, which is currently lacking in Public Health policy". 

In general, the manuscript is well written, but needs improvement, at least for the Discussion section, where the authors should make it clearer what differentiates the results of their study compared to previous studies and what new results were in their study.

Some other comments and suggestions:

Line 25 Specify in Abstract what Crl of RR means (Credible Intervals of Rate Ratios)

Lines 174-176 You can remove these 2 sentences "The number ... ring at 0". They can be found as explanations in Fig. 2 lines 211-213

Lines 221-223 Specify the cause

Lines 234 Explain why "even though these differences were not statistically significant"

For Table 2 and Table 3, from Lines 235-239, explain more clearly the differentiation results for these 4 clusters

Lines 361-362 The paper (Befort, Niaman and Perri, 2012; Robertson et al., 2018) is not found in References. Write this paper in References and assign a number for this paper.

Similar situation for Lines 364-365 (Heraclides, Kolokotroni and Charalambous, 2015)

Reviewer 2 Report

This is a well-written paper providing information on the association between geodemographic are classification and mortality in Cyprus. Data on Cyprus are lacking in this kind of research and therefore this study gives the reader valuable information. 

I have only minor comments, mostly regarding the form than content of the study.

1) Table 1 (page 4): Please correct the definition of the demographic indicator "age 0–14". In the definition, the authors write "population aged 0–4 years" - this is clearly a spelling mistake

2) Figure 1: Please remove the large heading of the figure "Four clusters". If the authors need to specify four clusters for the figure, it is better to use a note or include this in the title of Figure 1 below the figure.

3) Figure 3: Please remove the large heading of the figure "Smoothed Rate Ratios". This information should be given in the title of the figure 3 below the figure

4) page 8, lines 230–231: The authors wrote "While rates of premature mortality were higher in traditional-life rural communities, these communities appear to have 12% lower mortality rates compared to metropolitan areas." I would suggest to include in the sentence above that the 12% lower mortality rates were estimated for the adults aged 65 years and over. 

Reviewer 3 Report

Thank you to the authors for this interesting and well-written paper. I think that it definitely has the potential for publication. However, I think that there are some changes that should be made to the manuscript before publication. Some are quite specific, and some are more general. 

Firstly, there are a number of spelling/grammar mistakes throughout the manuscript. I haven’t got the time or inclination to go through all of them, but a few of them are detailed below:

Line 13: Comma after monitoring

Line 22: Reword sentence starting ‘Most defining’, it currently doesn’t make much sense to me!

Line 36: Change the wording to ‘Different degrees’

Line 44: Reword sentence, it's currently a bit confused

Line 53: Get rid of etc (shouldn't really be in an academic article)

If I were the authors I would proof read the article again.

The comments below are not to do with spelling/grammar:

Line 135: Did you try models where you didn’t normalise the data? It would be interesting to know if the results would have changed at all.

Line 160: Where did you get your prior distributions for the Bayesian analysis from? You don’t mention them at all in the manuscript and I think that it is important that you do include them.

Line 201: Are the radial plots the best way to present this data? I personally haven’t ever seen these in a journal article. Would a normal table be easier to read?

General Comment: Can you comment on any of the potential mechanisms between your calculated measures of socioeconomic status and mortality levels in Cyprus? Perhaps you could include something about this in the discussion?

General Comment: Will this data be used to inform the geographical allocation of health care resources in Cyprus in the future? If so, perhaps you could elaborate on this a bit in the discussion/conclusion?  

General Comment: I’m a little bit confused by the modelling strategy - why did you calculate the SMRs? It reads to me that you didn’t end up using them in the econometric analysis and instead used the crude mortality rates with the expected number of deaths as the offsetting variable in your Poisson model? If this is the case, you should delete the part about the calculation of the SMRs to reduce confusion.   

General Comment: It would be useful to see the mean values presented alongside the median values throughout (including the supplementary appendices)

Round 2

Reviewer 1 Report

The manuscript can be published in present form.